# Physical Activity during Preconception Impacts Some Maternal Outcomes—A Cross-Sectional Study on a Population of Polish Women

**DOI:** 10.3390/ijerph20043581

**Published:** 2023-02-17

**Authors:** Adrian Kruszewski, Paulina Przybysz, Joanna Kacperczyk-Bartnik, Agnieszka Dobrowolska-Redo, Ewa Romejko-Wolniewicz

**Affiliations:** 1Students’ Scientific Group Affiliated to 2nd Department of Obstetrics and Gynecology, Medical University of Warsaw, 00-315 Warsaw, Poland; 22nd Department of Obstetrics and Gynecology, Medical University of Warsaw, 00-315 Warsaw, Poland

**Keywords:** physical activity, pregnancy outcome, gestational diabetes mellitus

## Abstract

Background: Physical activity is an element of a healthy lifestyle and is safe in most pregnancies. The aim of this study was to assess the impact of physical activity levels before and during pregnancy on pregnancy outcomes for both the mother and child. Methods: A cross-sectional survey was conducted on a population of Polish women. An anonymous questionnaire was distributed electronically via maternity and parental Facebook groups. Results: The final research group included 961 women. The analysis showed that physical activity 6 months before pregnancy was associated with a lower risk of gestational diabetes mellitus (GDM), but physical activity during pregnancy showed no such association. In all, 37.8% of women with low activity in the first trimester, in comparison to 29.4% of adequately active women, gained an excessive amount of weight during pregnancy (*p* = 0.0306). The results showed no association between activity level and pregnancy duration, type of delivery or newborn birth weight. Conclusions: Our study indicates that physical activity during the preconception period is crucial to GDM occurrence.

## 1. Introduction

Physical activity is a crucial element of human health at every stage of life, including during pregnancy. Regular physical activity maintains and improves the cardiovascular and respiratory systems and reduces the risk of obesity. The 2018 update to the U.S. Department of Health and Human Services Physical Activity Guidelines advises that women who were physically active before pregnancy should continue these activities during pregnancy and the postpartum period [1]. The American College of Obstetricians and Gynecologists (ACOG) recommends 20–30 min of moderate-intensity exercise per day on most days of the week [2]. Women in uncomplicated pregnancies with no contraindications to physical activity can perform aerobic and strength conditioning exercises. The benefits of physical exercise during pregnancy for both the mother and the foetus have been reported in the literature. The ACOG recommendation suggests a possible relationship between physical activity and the risk of gestational diabetes mellitus (GDM). Physical inactivity has been recognised as an independent risk factor for maternal obesity and related pregnancy complications, including GDM.

In Poland, the incidence of GDM among pregnant women was estimated to be 4.7% in 2010, 6.9% in 2011 and 7.5% in 2012, with a dynamic upward trend [3]. GDM risk factors are well known; they include unmodifiable factors such as increasing maternal age, family history of diabetes, history of delivering a child with birth weight above 4000 g and previous stillbirth. GDM occurrence is also influenced by modifiable factors, including overweight and obesity. GDM is associated with numerous maternal and foetal complications. The most common complications are maternal hypertension, polyhydramnios, preterm labour, foetal macrosomia, perinatal injuries and intrauterine foetal death. After the delivery, blood glucose levels normalise in the majority of women. However, a history of GDM is a risk factor for type 2 diabetes mellitus (DM2) [4,5]. These women have a 4- to 10-fold increased risk of developing DM2 during their lifetime.

It has been reported in the literature that structured physical exercise during pregnancy reduces the risk of caesarean section by 15% [6]. Today, caesarean section is the most common obstetric surgery. Nevertheless, it should be remembered that caesarean section is still an operation with serious possible complications for both the mother and the foetus. Caesarean section is associated with a 10-fold higher risk of postpartum haemorrhage and thromboembolic complications, which are the leading causes of postpartum death in women in comparison to vaginal delivery [7,8]. In Poland, the caesarean section percentage is one of the highest in Europe, amounting to 43.85% [9].

The study aimed to verify whether physical activity before and during pregnancy can decrease GDM and caesarean section risk in a population of Polish women.

## 2. Materials and Methods

This was a cross-sectional survey. The analysis involved a sample of women after delivery. The inclusion criteria were the following: correctly completed questionnaire, singleton delivery and no contraindications to physical activity during pregnancy. Additionally, for the analysis of the impact of physical activity on GDM occurrence, participants with diabetes mellitus diagnosed before pregnancy (pregestational diabetes mellitus, PGDM) were excluded (Figure 1). The endpoints included the presence of GDM, weight gain during pregnancy, pregnancy duration and type of delivery. In the analysis of the effect of maternal physical activity on the newborn, the endpoint was the newborn’s birth weight.

The survey was self-composed by the authors, and it was prepared in the Polish language. The anonymous questionnaire consisted of single-choice closed, multiple-choice closed and open questions. The survey was voluntary, and it was distributed electronically using Google Forms (Google LLC, Mountain View, CA, USA). The questionnaire is presented in the Appendix A.

The data were collected from December 2021 to January 2022. The questionnaire was distributed via randomly chosen Polish maternity and parental Facebook groups. The final group included 961 women. Five women had diabetes mellitus diagnosed before pregnancy and were excluded from the analysis of the impact of physical activity on GDM occurrence. Respondents were divided into two groups for the purpose of the analysis: less physically active and adequately physically active. Women who were active less than 30 min per week or active 2–3 times for 30 min per week, which corresponds to, at most, 90 min per week, were defined as “less physically active”. Participants who were active 4–5 times for 30 min or more than 5 times for 30 min per week, which corresponds to more than 90 min per week, composed the group of “adequately physically active”.

Prepregnancy body mass index (BMI) was calculated by dividing prepregnancy weight in kilograms by height in metres squared. According to WHO criteria, the following ranges were defined: underweight: <18.5 kg/m^2^; normal range: 18.5–24.9 kg/m^2^; overweight: 25.0–29.9 kg/m^2^; obese: >30.0 kg/m^2^. According to prepregnancy BMI, gestational weight gain in kilograms was estimated using the recommendations of the Institute of Medicine and the National Research Council of the National Academies (US) [10]. The women were assigned to one of the following categories: insufficient weight gain, adequate weight gain and excessive weight gain. Delivery before 37 completed weeks of gestation was considered a premature birth, after 42 completed weeks of gestation was defined as a post-term delivery and between the 37th and 41st weeks of gestation was considered a term delivery. The delivery methods comprised the vaginal delivery group, a caesarean section due to medical indications group and a caesarean section for undefined or unclear indications group. The vaginal delivery group included spontaneous delivery, forceps delivery and vacuum extractor delivery. The presence or absence of indications for caesarean section was determined by the analysis of answers to the open question about the reason for the caesarean section compared with the Recommendations of the Polish Society of Gynecologists and Obstetricians regarding caesarean sections [11]. Newborn birth weight was divided into three ranges: below 2500 g, between 2500 g and 4000 g and above 4000 g.

The data from the questionnaire were processed in Microsoft Excel (Microsoft Corporation, Redmond, WA, USA) and STATISTICA 13.3 (TIBCO Software Inc., Palo Alto, CA, USA). The data are reported as absolute numbers and percentages. The groups of less physically active and adequately physically active women were compared. The survey data were analysed using Fisher exact tests. *p*-values of <0.05 were considered statistically significant.

## 3. Results

Only 27% (n = 259) of women were adequately physically active during the 6 months before conception. The percentage of women in the adequately physically active group changed in each trimester and was the lowest at the end of pregnancy. In the first trimester, 20% (n = 197) of women were adequately physically active; in the second trimester, 24% (n = 227) were adequately physically active; and in the third trimester, 19% (n = 181) were adequately physically active. The main baseline characteristics of less physically active and adequately physically active women during the 6 months before pregnancy are presented in Table 1. Most of the respondents in both groups were women between 26 and 35 years old (median age, 30.5 years old). Compared with less physically active women, adequately physically active women were from a city (population > 500,000; *p* = 0.0002), with higher education (*p* = 0.0339) and normal prepregnancy body weight (*p* = 0.0024). In the group of obese women, the majority (89%) were women with low physical activity (*p* = 0.0013).

### 3.1. Gestational Diabetes Mellitus

For the analysis of the impact of physical activity before pregnancy and in each trimester on the risk of GDM occurrence, 956 women were included. Table 2 presents the relationship between physical activity and the occurrence of GDM. Physical activity during the 6 months before pregnancy had a major impact on the occurrence of GDM. Gestational diabetes mellitus occurred in 14.1% of less physically active women and in only 7.0% of adequately physically active women during this period (*p* = 0.0025). There was no association between exercise during pregnancy and the risk of GDM. Physical activity in any trimester did not reduce the risk of gestational diabetes mellitus.

The results regarding the relationship between physical activity and gestational weight gain, pregnancy duration, the type of delivery and newborn birth weight are presented in Table 3.

### 3.2. Weight Gain during Pregnancy

Significantly more women from the less physically active group during the 6 months before pregnancy (38.0% vs. 30.9%; *p* = 0.0413) and in the first trimester (37.8% vs. 29.4%; *p* = 0.0306) gained an excessive amount of weight during pregnancy. Furthermore, most adequately physically active women did not gain enough weight during pregnancy. Insufficient weight gain was recorded in 36.3% of adequately physically active women before pregnancy and in 40.1% of adequately physically active women in the first trimester, in comparison to 29.9% and 29.5%, respectively, among less physically active women (*p* = 0.0611 and *p* = 0.0059).

### 3.3. Duration of Pregnancy

Post-term delivery was more common in women with low physical activity than in those with adequate activity during the 6 months before pregnancy. Post-term deliveries occurred in 7.0% of the women who were less physically active before pregnancy and in 3.1% of the women who were adequately physically active before pregnancy (*p* = 0.0213). There was no association between physical activity during any trimester of pregnancy and the pregnancy duration.

### 3.4. Type of Delivery

No statistically significant association between the type of delivery and physical activity during the 6 months before pregnancy was observed. Caesarean sections due to medical indications were performed at the same frequency in both groups. However, adequate physical activity during the first trimester of pregnancy was associated with a lower risk of caesarean section. Caesarean section due to medical indications was more common in women in the less physically active group than in the adequately physically active group (37.6% vs. 29.4%; *p* = 0.0373).

### 3.5. Newborn Birth Weight

The frequency of physical activity before conception and during pregnancy was not associated with newborn birth weight. Newborns with both low and high birth weights were born with similar frequency in both groups.

### 3.6. Medical Advice for Pregnant Women

Regarding the question of whether they had received advice during their pregnancy from a qualified healthcare professional (doctor/midwife) about optimal pregnancy-related weight gain, half of the respondents (50.3–52.3%) answered that they had not received any advice. Concerning the question of whether they had received advice during their pregnancy from a qualified healthcare professional regarding the recommended physical activity, 63.7% (612) of respondents answered that they had not received any advice about it. In the group of women that had received advice about physical activity, more women were adequately physically active in comparison to the group of women that had not received that advice (*p* = 0.0002).

## 4. Discussion

The results of our research show that GDM is more common in women who were less physically active during the 6 months before pregnancy. Research conducted by Dempsey et al. showed that women who participated in any recreational physical activity during the year before pregnancy experienced a 66 per cent reduction in risk of gestational diabetes mellitus [12]. Similar results were presented by Zhang et al. [13]. Their analysis included 21,765 women, and the results showed that prepregnancy physical activity—in particular, increasing vigorous physical activity—was associated with a significantly lower risk of GDM.

Studies concerning the impact of physical activity levels during pregnancy on the incidence of GDM have shown different results. A study conducted by Ehrlich et al. [14] suggested that meeting current recommendations for exercise during the first trimester of pregnancy does not confer reductions in the risks of abnormal screening and GDM. On the other hand, compared with less vigorous activities, exercise intensity that reaches at least 60% of heart rate reserves during pregnancy while gradually increasing physical activity energy expenditure reduces the risk of gestational diabetes mellitus [15]. Oken et al. conducted a study on over 1800 women, and they found that women who engaged in vigorous physical activity before pregnancy and light-to-moderate or vigorous activity during pregnancy had a reduced risk of developing gestational diabetes mellitus [16]. In our analysis, physical activity level during pregnancy was not associated with the incidence of GDM. A systematic review and meta-analysis conducted by Mijatovic-Vukas et al. showed that physical activity is a promising intervention for the prevention of GDM, and engaging in any physical activity, even below the guideline level, suggested a protective association with GDM risk [17].

The present study revealed that women who were adequately physically active during the first trimester of pregnancy had a higher risk of insufficient gestational weight gain, according to the recommendations. We noticed that low physical activity during the previous 6 months and in the first trimester of pregnancy was associated with excessive gestational weight gain. Experts indicate that gestational weight gain is modifiable through diet and physical activity [18]. Barakat et al. conducted a randomised controlled trial wherein pregnant women in the intervention group received standard care and all aspects of a structured and supervised moderate exercise intervention programme, and pregnant women allocated to the control received standard care. The main finding of this review was that the exercise programme reduced the total (mean) maternal weight gain, as well as the cases of excessive weight gain and GDM [19]. However, in a study by Ruifrok et al., it was shown that neither physical activity nor sedentary behaviour was associated with gestational weight gain [20]. A systematic review of interventional trials conducted by Streuling suggested that physical activity during pregnancy might be a successful strategy in restricting gestational weight gain [21].

Our study revealed that physical activity below 90 min per week in the period 6 months before pregnancy was associated with a higher risk of post-term delivery. However, the physical activity level during pregnancy did not appear to impact pregnancy duration. Similar results were presented by Cavalli et al., who found that maternal leisure-time physical activity during pregnancy did not increase the risk of preterm delivery [22]. Hatch et al., in their research, found no adverse effect of maternal exercise during pregnancy on gestational length among live births [23]. On the other hand, Evenson et al. conducted research that showed that vigorous leisure activity during the first and second trimesters of pregnancy was related to a decreased risk of preterm birth, with the relationship being stronger for the second trimester; there was no association with post-term birth incidence [24]. In a paper by Straughen et al. involving 1410 African American women, it was shown that women who participate in leisure-time physical activity during pregnancy have a decreased prevalence of preterm delivery [25].

The results of our study showed no relationship between the occurrence of vaginal delivery and physical activity level during the 6 months before pregnancy or during pregnancy. Caesarean section rates are progressively rising in many parts of the world. One suggested reason for this is increasing requests by women for a caesarean section in the absence of clear medical indications [26]. The results from research by Chen et al. indicate that increasing physical activity in the first two trimesters and decreasing sedentary time in the third trimester could be helpful in increasing the chances of vaginal delivery [27]. In a study by Russo et al. involving 1313 Hispanic women, it was shown that sedentary activity increased the risk of caesarean delivery, and moderate-intensity and household/caregiving physical activity reduced the odds of unplanned caesarean deliveries [28]. A meta-analysis from Domenjoz et al. revealed that structured aerobic or resistance exercise programmes during pregnancy decreased the risk of caesarean delivery by 15% [6]. In our analysis, adequate physical activity (over 90 min per week) in the first trimester was associated with a lower risk of caesarean section due to medical indications, but the same level of activity in the third trimester was associated with a higher risk of caesarean section performed due to undefined indications.

Experts point to the fact that low birth weight and macrosomia are associated with short- and long-term health consequences. A systematic review and meta-analysis of observational studies suggest an inverted U-shaped association between physical activity and birth weight: moderate levels seem to be associated with an increased birth weight, while high levels of physical activity seem to be associated with a decreased birth weight [29]. In the present review, we found that there were no statistically significant differences in newborn birth weight between less and more physically active women.

According to a paper by Santo et al., the majority of pregnant women reported receiving physical activity advice in the context of prenatal care, but at least 25% appeared not to be receiving such advice, particularly multiparous and obese women [30]. A study by Ferrari et al. showed that women commonly reported receiving physical activity advice that was conservative [31]. In our research, approximately 48% of pregnant women received advice from a qualified healthcare professional about optimal gestational weight gain, and only 36% of participants were advised regarding the recommended physical activity during pregnancy. In a study by Findley et al., it was shown that most participants wanted advice that was evidence-based or was from professionals who were trained to know what constitutes safe activity while pregnant [32]. In the present study, pregnant women most often undertook activities such as walking, marching and home gymnastics. The ACOG guidelines reveal that, in general, participation in a wide range of recreational activities appears to be safe [33].

There are several limitations of our survey. The questionnaire was self-composed by the authors and had not been previously validated in a pregnant population. We could not verify all the answers. Some questions were subjective, e.g., women may be led to over- or underestimate their physical activity level. In our study, the level of physical activity was determined by its duration without taking into account the level of activity, which may be a limitation. These limitations might affect the reliability and validity of the questionnaire. The major advantages of our research were the large research group and the fact that 98.7% of the questionnaires were completed correctly. Taking all this into account allowed us to draw reasonable conclusions.

## 5. Conclusions

Our study indicates a beneficial role of physical activity during the preconception period in decreasing the risk of GDM occurrence. Indications for caesarean section were less likely in pregnant women who were adequately active in the first trimester of pregnancy. The levels of activity in the first trimester of pregnancy and during the 6 months prior had an impact on gestational weight gain. The findings prove that the offspring of a physically active mother is not at risk of prematurity or low birth weight. It seems important to increase the level of information and education about physical activity provided by healthcare professionals to future mothers.

## Figures and Tables

**Figure 1 ijerph-20-03581-f001:**
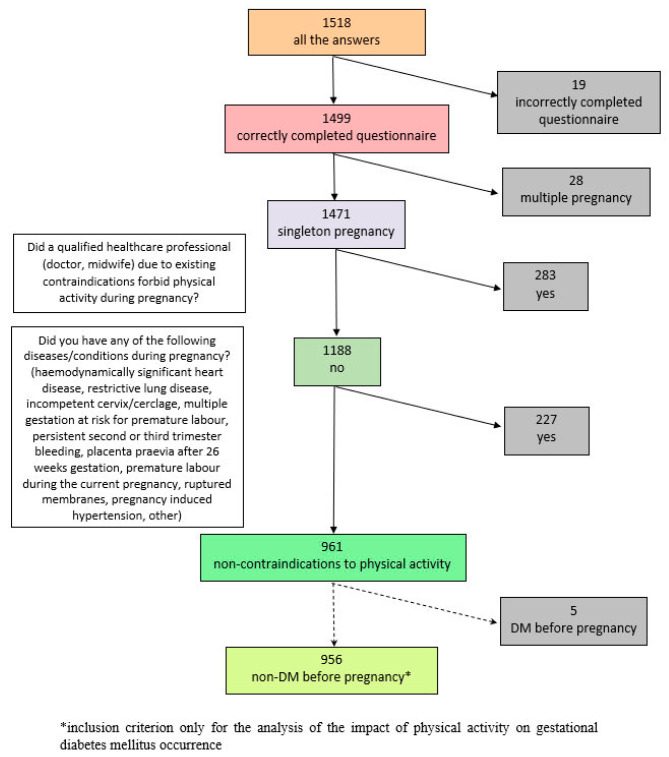
The inclusion criteria.

**Table 1 ijerph-20-03581-t001:** Characteristics of the study group and subgroups according to the level of physical activity in the period of 6 months before pregnancy.

	Study Group N = 961	Less Physically Active N = 702	Adequately Physically Active N = 259	*p*
	% (N)	% (N)	% (N)	
Age (years)				
16–20	2.2 (21)	2.3 (16)	1.9 (5)	1.00
21–25	14.2 (136)	15.1 (106)	11.6 (30)	0.18
26–30	37.8 (363)	37.2 (261)	39.4 (102)	0.55
31–35	33.9 (326)	32.9 (231)	36.7 (95)	0.28
≥36	12.0 (115)	12.5 (88)	10.4 (27)	0.43
Habitation				
Countryside	27.6 (265)	30.3 (213)	20.1 (52)	0.0015
Town <50,000	21.2 (204)	22.4 (157)	18.1 (47)	0.18
City 50,000–100,000	10.9 (105)	11.4 (80)	9.7 (25)	0.49
City 100,000–500,000	11.9 (114)	10.8 (76)	14.7 (38)	0.12
City >500,000	28.4 (273)	25.1 (176)	37.5 (97)	0.0002
Education				
Primary	0.8 (8)	1.0 (7)	0.4 (1)	0.69
Vocational	3.6 (35)	4.3 (30)	1.9 (5)	0.12
Secondary	19.7 (189)	20.7 (145)	17.0 (44)	0.23
Higher	75.9 (729)	74.1 (520)	80.7 (209)	0.0339
Prepregnancy BMI (kg/m^2^)				
Underweight (<18.5)	6.0 (58)	6.1 (43)	5.8 (15)	1.00
Normal (18.5–24.9)	70.7 (679)	67.9 (477)	78.0 (202)	0.0024
Overweight (25–29.9)	15.9 (153)	17.0 (119)	13.1 (34)	0.16
Obese (>30)	7.4 (71)	9.0 (63)	3.1 (8)	0.0013
Pregnancy				
First	39.4 (379)	35.9 (252)	49.0 (127)	0.0003
Second	41.0 (394)	43.3 (304)	34.7 (90)	0.0180
Third	14.7 (141)	15.7 (110)	12.0 (31)	0.18
Fourth or another	4.9 (47)	5.1 (36)	4.2 (11)	0.74

**Table 2 ijerph-20-03581-t002:** Presence of GDM according to the level of physical activity during the 6 months before pregnancy and in each trimester.

	Less Physically Active	Adequately Physically Active	
	% (N)	% (N)	*p*
During 6 months before pregnancy			
Non-GDM	85.9 (601)	93.0 (238)	0.0025
GDM	14.1 (99)	7.0 (18)	
First trimester			
Non-GDM	87.0 (662)	90.8 (177)	0.18
GDM	13.0 (99)	9.2 (18)	
Second trimester			
Non-GDM	87.3 (638)	89.3 (201)	0.49
GDM	12.7 (93)	10.7 (24)	
Third trimester			
Non-GDM	87.6 (680)	88.3 (159)	0.90
GDM	12.4 (96)	11.7 (21)	

**Table 3 ijerph-20-03581-t003:** Gestational weight gain, pregnancy duration, the type of delivery and newborn birth weight according to the level of physical activity during the 6 months before pregnancy and in each trimester.

	Less Physically Active	Adequately Physically Active	
	% (N)	% (N)	*p*
**Weight gain during pregnancy**			
During 6 months before pregnancy			
Insufficient weight gain	29.9 (210)	36.3 (94)	0.0611
Adequate weight gain	32.1 (225)	32.8 (85)	0.82
Excessive weight gain	38.0 (267)	30.9 (80)	0.0413
First trimester			
Insufficient weight gain	29.5 (225)	40.1 (79)	0.0059
Adequate weight gain	32.7 (250)	30.5 (60)	0.61
Excessive weight gain	37.8 (289)	29.4 (58)	0.0306
Second trimester			
Insufficient weight gain	30.7 (225)	34.8 (79)	0.25
Adequate weight gain	32.0 (235)	33.0 (75)	0.81
Excessive weight gain	37.3 (274)	32.3 (73)	0.18
Third trimester			
Insufficient weight gain	30.6 (239)	35.9 (65)	0.18
Adequate weight gain	32.4 (253)	31.5 (57)	0.86
Excessive weight gain	37.0 (288)	32.6 (59)	0.30
**Pregnancy duration**			
During 6 months before pregnancy			
Premature birth	4.4 (31)	4.6 (12)	0.86
Term delivery	88.6 (622)	92.3 (239)	0.12
Post-term delivery	7.0 (49)	3.1 (8)	0.0213
First trimester			
Premature birth	4.3 (33)	5.1 (10)	0.70
Term delivery	89.4 (683)	90.3 (178)	0.80
Post-term delivery	6.3 (48)	4.6 (9)	0.50
Second trimester			
Premature birth	4.5 (33)	4.4 (10)	1.00
Term delivery	89.5 (657)	89.9 (204)	1.00
Post-term delivery	6.0 (44)	5.7 (13)	1.00
Third trimester			
Premature birth	4.2 (33)	5.5 (10)	0.43
Term delivery	89.5 (698)	90.1 (163)	0.89
Post-term delivery	6.3 (49)	4.4 (8)	0.39
**Type of delivery**			
During 6 months before pregnancy			
Vaginal delivery	60.0 (421)	60.6 (157)	0.88
Caesarean section (medical indications)	36.3 (255)	34.8 (90)	0.70
Caesarean section (undefined indications)	3.7 (26)	4.6 (12)	0.58
First trimester			
Vaginal delivery	58.9 (450)	65.0 (128)	0.12
Caesarean section (medical indications)	37.6 (287)	29.4 (58)	0.0373
Caesarean section (undefined indications)	3.5 (27)	5.6 (11)	0.22
Second trimester			
Vaginal delivery	60.4 (443)	59.5 (135)	0.82
Caesarean section (medical indications)	36.2 (266)	34.8 (79)	0.75
Caesarean section (undefined indications)	3.4 (25)	5.7 (13)	0.12
Third trimester			
Vaginal delivery	59.7 (466)	61.9 (112)	0.61
Caesarean section (medical indications)	37.3 (291)	29.8 (54)	0.07
Caesarean section (undefined indications)	3.0 (23)	8.3 (15)	0.0023
**Newborn birth weight**			
During 6 months before pregnancy			
Birth weight <2500 g	4.3 (30)	5.4 (14)	0.49
Birth weight 2500 g–4000 g	84.3 (592)	84.6 (219)	1.00
Birth weight >4000 g	11.4 (80)	10.0 (26)	0.64
First trimester			
Birth weight <2500 g	4.2 (32)	6.1 (12)	0.25
Birth weight 2500 g–4000 g	84.8 (648)	82.7 (163)	0.51
Birth weight >4000 g	11.0 (84)	11.2 (22)	1.00
Second trimester			
Birth weight <2500 g	4.5 (33)	4.8 (11)	0.86
Birth weight 2500 g–4000 g	84.3 (619)	84.6 (192)	1.00
Birth weight >4000 g	11.2 (82)	10.6 (24)	0.90
Third trimester			
Birth weight <2500 g	4.4 (34)	5.5 (10)	0.55
Birth weight 2500 g–4000 g	84.5 (659)	84.0 (152)	1.00
Birth weight >4000 g	11.1 (87)	10.5 (19)	0.90

## Data Availability

Data are available on request.

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
