# Peer review of "Physical Activity during Preconception Impacts Some Maternal Outcomes—A Cross-Sectional Study on a Population of Polish Women"

_ijerph, 2023, doi:10.3390/ijerph20043581_

Round 1

Reviewer 1 Report

Title: Physical Activity during Preconception, not only during Pregnancy, Impacts Some Maternal Outcomes – A Cross Sectional Study on the Population of Polish Women

Manuscript ID: ijerph-2106624

Higher physical activity during preconception and pregnancy has been associated with a reduced risk of several maternal health outcomes in short and long periods. The topic is worth researching in a particular population with possible different norms and lifestyle regarding pregnant women.  However, I have some comments regarding the manuscript:

·       The title is wrong as the study found no association of physical activity during pregnancy with the selected maternal outcomes.

·      Despite some advantages, a self-report questionnaire has some major drawbacks that might affect the reliability and validity of the questionnaire especially with respect to maternal and neonatal outcomes (endpoints).

·      How did the authors define physical activity? Was it based on only duration of the activities reported by the participants? If so, maybe shorter duration like less than 90 minutes was intensive activity and it therefore, couldn’t be categorized as “less active”.

·      There are a number of English errors throughout the text e.g. “stillborn” which is “stillbirth” in the correct form.

·      Is “PGDM” the abbreviation for participants with diabetes mellitus diagnosed before pregnancy? If so, it is not GDM and it should be considered as pre-existing diabetes (either type 1 or type 2 diabetes).

·      At Table 2, why the results of “PGDM” are included? They were excluded from the analysis as it was mentioned in the text. Which statistical test has been used for the analyses in Table 2?

Author Response

Response to Reviewer 1 Comments

Point 1: The title is wrong as the study found no association of physical activity during pregnancy with the selected maternal outcomes.

Response 1: As suggested, we decided to change the title. The new title is “Physical Activity during Preconception Impacts Some Maternal Outcomes – A Cross Sectional Study on the Population of Polish Women”.

Point 2: Despite some advantages, a self-report questionnaire has some major drawbacks that might affect the reliability and validity of the questionnaire especially with respect to maternal and neonatal outcomes (endpoints).

Response 2: On page 9, in Discussion section, the last paragraph says about the limitations of our study. It mentions that we could not verify all the answers. In this paragraph, we added sentences about the limitations of the study.

Point 3: How did the authors define physical activity? Was it based on only duration of the activities reported by the participants? If so, maybe shorter duration like less than 90 minutes was intensive activity and it therefore, couldn’t be categorized as “less active”.

Response 3: In the questionnaire we did not provided a definition of physical activity. In the study, we asked women to declare the duration of activity per week from the following options: less than 30 minutes per week; 2-3 times for 30 minutes per week; 4-5 times for 30 minutes per week; more than 5 times for 30 minutes per week. This question was subjective and women may have been led to both over and underestimating their physical activity level. In the last paragraph in Discussion section we added sentences about the limitations of the study.

Point 4: There are a number of English errors throughout the text e.g. “stillborn” which is “stillbirth” in the correct form.

Response 4: The text has undergone language editing.

Point 5: Is “PGDM” the abbreviation for participants with diabetes mellitus diagnosed before pregnancy? If so, it is not GDM and it should be considered as pre-existing diabetes (either type 1 or type 2 diabetes).

Response 5: The abbreviation "PGDM" was used for women declaring in the questionnaire that they had diabetes mellitus diagnosed in before pregnancy and it is of course not GDM, we agree.

Point 6:  At Table 2, why the results of “PGDM” are included? They were excluded from the analysis as it was mentioned in the text. Which statistical test has been used for the analyses in Table 2?

Response 6: We have made changes to Table 2 by removing "PGDM" from it. We used Fisher exact tests to analyze data in Table 2.

Reviewer 2 Report

It is a good and interesting study, worth reading. However, the following points should be taken care of:

1. Method section

a) Define exercise: For example, you mentioned in the introduction section exercise as the "American College of Obstetricians and Gynecologists (ACOG) recommends 20-30 minutes of moderate-intensity exercise per day on most days of the week". Either you consider the same i.e., 150 minutes of moderate-intensity exercise per week OR any other specific guideline (since there are no such specific pre-pregnancy exercise guidelines) you have followed in the current work. Merely mentioning in the introduction does not provide any clarity that you had used the same in your study as well.    

b) Please provide reason(s) for defining the category "adequately physical activity" with more than 90 minutes per week. 

c) Shorten the second paragraph in the method section (lines 75 - 91). Include only the important items and the rest should be presented as supplementary material. 

d) Since GDM is meant as the primary outcome of the study, a predictive analysis should be performed. Use the duration of exercise (in minutes), not the categories, against the dependent variable. 

2. Result

Please mention the weight/weight gain/body mass index in Tables 1 & 3 of the participants instead of the number and percentage of various categories. The number and percentage should be presented in parentheses. 

3. Discussion and Conclusion:

Both need modification after consideration of predictive analysis. 

Author Response

Response to Reviewer 2 Comments

  • Method section:

Point 1: Define exercise: For example, you mentioned in the introduction section exercise as the "American College of Obstetricians and Gynecologists (ACOG) recommends 20-30 minutes of moderate-intensity exercise per day on most days of the week". Either you consider the same i.e., 150 minutes of moderate-intensity exercise per week OR any other specific guideline (since there are no such specific pre-pregnancy exercise guidelines) you have followed in the current work. Merely mentioning in the introduction does not provide any clarity that you had used the same in your study as well.

Response 1:  We are sorry, but there are no adequate Polish recommendations about physical activity during pregnancy, so in our study we followed ACOG recommendation, which we mentioned in the introduction. “20-30 minutes of (…) exercise per day on most days of the week”, we interpreted this sentence as: most days, i.e. at least 4 days a week for a minimum of 20-30 minutes, which gives minimum 80-120 minutes a week. We chose to set the cut-off point at 90 minutes per week because in our self-composed questionnaire we asked women to declare the duration of activity per week from the following options: less than 30 minutes per week; 2-3 times for 30 minutes per week; 4-5 times for 30 minutes per week; more than 5 times for 30 minutes per week. We described the process of division into two groups: less physically active and adequate physically active in Materials and Methods section (lines 97-103).

Point 2: Please provide reason(s) for defining the category "adequately physical activity" with more than 90 minutes per week.

Response 2: We defined the category "adequately physical activity" with more than 90 minutes per week for at least two reasons. First of all, 90 minutes per week is minimum time recommended by ACOG (at least 4 days per 20-30 minutes). Second, our survey included questions about physical activity in multiples of 30 minutes.

Point 3: Shorten the second paragraph in the method section (lines 75 - 91). Include only the important items and the rest should be presented as supplementary material.

Response 3: The second paragraph has been shortened.

Point 4: Since GDM is meant as the primary outcome of the study, a predictive analysis should be performed. Use the duration of exercise (in minutes), not the categories, against the dependent variable.

Response 4: The study was designed in a way that only allows us to obtain information about the amount of exercise in multiples of 30 minutes. Therefore, the research group was divided into two groups: women exercising less than 90 minutes a week and exercising more than 90 minutes a week. The statistical analysis was based on this division. We do not have information about duration of exercise in minutes and probably it wouldn’t be possible to get such information from patients. Usually someone realizes that is walking more than half an hour, but not how many minutes. On the other hand, we do not think that the number of minutes impacts the GDM risk, we think that the number of hours spent on physical activity play a role, that is why we decided to distinguish less active and adequately active. Anyway, thank You for Your suggestion, such study could be projected with new technological equipment counting the exact time of exercise.

  • Result:

Point 5: Please mention the weight/weight gain/body mass index in Tables 1 & 3 of the participants instead of the number and percentage of various categories. The number and percentage should be presented in parentheses.

Response 5: In Table 1, information on the definition of body mass index ranges has been added. In Table 3 the weight gain was estimated using the recommendations of the Institute of Medicine and National Research Council of the National Academies (US). Recommended weight gain during pregnancy depends on the pre-pregnancy BMI. We used a division into categories: too low, adequate and too much weight gain because according to recommendations there is not a universal value (in kilograms) for all women.

  • Discussion and Conclusion:

Point 6: Both need modification after consideration of predictive analysis.

Response 6: Modifications have been made.

Round 2

Reviewer 1 Report

Thank the authors for the revision of the manuscript. Although the authors attempted to revise the manuscript, according to my comments, there are still some comments I would like to mention:  

·      There are quite number of grammatical and editorial mistakes throughout the manuscript. Therefore, editing of English language is necessarily required for the manuscript to ensure consistency and accuracy in grammar, punctuation, and etc… 

·    The “PGDM” abbreviation should be defined for the first time. As shown in Fig 1, these are women with preexisting DM and not GDM.

·   “CC” is wrong abbreviation for cesarean section.

·   Considering only the duration and not the intensity of the physical activities by individuals may lead to invalid results and therefore, the findings of the study should be taken with caution.

Author Response

Point 1: There are quite number of grammatical and editorial mistakes throughout the manuscript. Therefore, editing of English language is necessarily required for the manuscript to ensure consistency and accuracy in grammar, punctuation, and etc…

Response 1: The manuscript has undergone English language editing by MDPI.

Point 2: The “PGDM” abbreviation should be defined for the first time. As shown in Fig 1, these are women with preexisting DM and not GDM.

Response 2: The abbreviation was defined in lines 71-72.

Point 3: “CC” is wrong abbreviation for cesarean section.

Response 3: We have removed this abbreviation from the manuscript

Point 4: Considering only the duration and not the intensity of the physical activities by individuals may lead to invalid results and therefore, the findings of the study should be taken with caution

Response 4: Thank You for Your comments. We are aware of some limitations of our study and we mention them in the last paragraph of the Discussion. In the questionnaire, we asked women about the type of activity they undertake. They most often undertook activities such as walking, marching, and home gymnastics. Based on this, we suspect that it was not intense physical activity. Your review helped us improve the manuscript.

Reviewer 2 Report

Well done

Author Response

Point 1: Well done

Response 1: Thank You for Your comments. Your review helped us improve the manuscript.
